# Environmental influences on evolvable robots

Karine Miras *, Eliseo Ferrante , A. E. Eiben

Computer Science Department/Computational Intelligence Group Vrije Universiteit Amsterdam, Amsterdam, Netherlands

* k.dasilvamirasdearaujo@vu.nl

## Abstract

The field of Evolutionary Robotics addresses the challenge of automatically designing robotic systems. Furthermore, the field can also support biological investigations related to evolution. In this paper, we evolve (simulated) modular robots under diverse environmental conditions and analyze the influences that these conditions have on the evolved morphologies, controllers, and behavior. To this end, we introduce a set of morphological, controller, and behavioral descriptors that together span a multi-dimensional trait space. Using these descriptors, we demonstrate how changes in environmental conditions induce different levels of differentiation in this trait space. Our main goal is to gain deeper insights into the effect of the environment on a robotic evolutionary process.

**Data Availability Statement:** All relevant data are available in the Supporting Information and the Github repository at https://github.com/ci-group/revolve/tree/a0a6496812cbec1208c3eb9fa4a0a21598ecb732/

## 1 Introduction

Natural evolution has inspired computer science to develop a digital counterpart, Evolutionary Computing, applicable to solving optimization, design, and modeling problems [1]. Evolutionary Computing can be considered as a 20th-century incarnation of the Darwinian principles *in silico*. Recent advances in robotics, rapid prototyping (3D-printing), and material science are opening up the potentials for the Evolution of Things—a 21st-century version *in materio* [2, 3].

Applying evolutionary methods to robots leads to a new kind of artificial evolution, one where the individuals are embodied, situated, and actuated. This is different from Evolutionary Computing where the population members are not embedded in time and space; they do not 'do' anything, but only represent a candidate solution of a search problem.

Evolutionary Robotics is concerned with evolving the morphologies (bodies) or controllers (brains), or both for simulated or real robots [4]. Notably, the interactions between the environment and the phenotypes of robots are an important factor in the evolutionary process, and this holds in a real-world setting as well as in simulations where physical interactions are simulated by some physics engine. The fitness of a robot is then determined by three factors, the body, and the brain, and the environment. Hence, by deduction, the result of the whole robotic evolutionary process depends on the environment as well. Taking a biological perspective, we can note that in natural evolution the environment largely determines the evolved life-forms [5–7]. The concept of Convergent Evolution is related to phenotypic convergence observed in nature as evidence that similar ecological conditions might select for analogous

experiments/Environmental_influences_on_
evolvable_robots.

**Funding:** The author(s) received no specific
funding for this work.

**Competing interests:** The authors have declared
that no competing interests exist.

evolutionary solutions in species with different genotypical ancestors [8]. This way, totally unrelated species evolved in similar environments can present very similar traits. "If environmental conditions favor one phenotype, then populations may diverge phenotypically and genetically through local adaptation."([9], as cited in [10]) For instance, "strawberry poison dart frogs are highly polymorphic, and genetic distances among populations are more strongly associated with phenotypic differences than with geographic distances, suggesting a role for local adaptation related to predation and aposematism."([11], as cited in [10]). Therefore, there are solid reasons to expect that the result of a robotic evolution process, body and brain, and thus behavior, will depend on the environment.

The existing literature on this subject can be divided in two categories, one where the robot bodies are fixed and only the controllers evolve and one where both the robot bodies and controllers undergo evolution. Using fixed robot bodies, there is a limited number of studies showing that different environments might produce different brains. For example, in the context of collective systems, the study in [12] showed that a flat environment produced an individual brain that did not induce complex self-organized strategies, while an environment with slope produced complex division of labour [13], analogously to what can be observed in leaf-cutter ants [14]. Considering the other category we can note that experimental work with evolvable bodies is scarce and limited in scope; importantly, the lack of studies into the effects of the environment is even more severe. The usual approach is to fix an environment and a task and evolve robots for the given combination. Varying the environment and investigating the effects on the evolving robots, specifically on the evolving bodies has hardly been addressed. The only study we know about is that of [15].

In this paper we research the effect of the environment in morphologically evolving robot systems extending our former work. In [16] we have demonstrated a case of a drastic environmental change that did not induce any significant changes neither in the evolved shapes nor in the emergent behaviors of the robots. Here we extend our work in [16] in two aspects, 1) running evolution in dynamically changing environmental conditions, further to static ones, and 2) by considering properties of robot controllers –not only morphologies and behaviour. In particular, we investigate the following questions:

- How do environmental conditions determine behavioral properties?

- How do environmental conditions determine morphological properties?

- How do environmental conditions determine properties of the controller?

The main contributions of this work are 1) explicitly putting the issue of the effect of the environment on the Evolutionary Robotics research agenda and noting that it is hard to find different environments that lead to different robots, 2) providing an open source simulator and test suite to facilitate further research, 3) new insights into environmental influences on evolved robot traits.

Let us note that it is hard to investigate environmental influences experimentally because it is very difficult to design environments that do induce phenotypic differentiation. That is to say, it is hard to design diverse environments that lead to diverse phenotypes. Notably, while we have experimented with multiple (types of) environments, the cases we present here are the only ones that led to differentiation. Curiously, in these unpublished experiments the same type of robots evolved in very different environments. We hypothesize that it has to do with the relative simplicity of these environments compared to the richness of (environmental) factors that determine the selection pressure in nature. Some of these unpublished results can be found in the supplementary material S1 Appendix.

Our approach to the above issues is experimental. We define quantitative descriptors of behavioral, morphological, and controller traits, specify environmental conditions and compare the evolutionary dynamics and the emerging populations in the trait space.

## 2 Related work

Evolutionary Robotics (ER) is an active field of research with significant achievements and many challenges [17, 18]. The conjoint evolution of robot morphologies and controllers was first explored by Sims' seminal work [19], but most of the ER studies concern the evolution of the robot controllers, and only a few look into the robot morphologies [20]. This bias is regrettable since it has been noted that advanced intelligence depends not only on the brain but also on the body [21–23]. Still, there are some interesting studies that, not only perform morphological evolution but even propose methods of morphological development, aiming to increase evolvability and robustness. The effect of different developmental mechanisms was studied in [24] by developing the stiffness of soft robots according to environmental changes, while a method for phenotypic plasticity of morphology and controller was proposed in [25]. Nevertheless, although such methods concern mechanisms regulated by the environment during lifetime, environmental changes were mostly caused by the displacement of the robot itself, resulting in differential sensing over time, therefore no actual 'changing' environment was considered, while robots evolved always in a flat plane. In [26] though, reconfigurable robots were evolved and managed to cope with actual changes in the environmental conditions as they moved about, but no quantification of this effect on the morphological level was provided. Moreover, the effect of diverse levels of gravity was investigated in [27], showing the emergence of different behaviors in these different environments, but again, providing no concrete measurements of differences in morphological traits. Finally, [15] an information-theoretic measurement of complexity was utilized to assess virtual creatures evolved in a vast range of environments. The authors demonstrated that increasing the complexity of the environmental conditions might result in an increase to the morphological complexity of the creatures. However, measuring complexity does not provide clear insights concerning properties of intelligible morphological traits, for instance, the number of limbs a robot has. Importantly, two environments could be equally complex, but induce the emergence of different phenotypic and behavioral traits.

In previous work [16, 28–31] we carried out a handful of investigations regarding the same robot framework utilized in this present paper. Firstly, we evolved robots that were "isolated" from any environmental influences, using morphological novelty search as a search criterion, which means the fitness function did not account for their performance on the task (for example locomotion on a flat terrain environment). In this analysis, we utilized a set of morphological descriptors that capture relevant properties of robot bodies and discovered a bias in the search space, that is, a tendency for determined morphological traits to be more often sampled by the reproduction operators.

This tendency is mainly characterized by robots that possess one or two limbs only, whereby other modules, that is, smallest morphology units, contribute to making these limbs a little longer. In summary, these robots presented an I-shape, similar to the shape of a "snake". To verify whether this bias would persist when environmental influences are introduced, we evolved robots using a behavior-oriented fitness function, and, curiously, the emergent robots presented this very same trait. Suspecting that the emergence of this trait was due to the bias, we realized new experiments evolving robots again for this same task and environment, this time rewarding not only locomotion, but also morphological diversity, and even explicitly rewarding the growth of limbs. Surprisingly, while these new populations that emerged to

locomote became indeed diverse (and multi-limb), the robots presented a worse performance on the task. These results suggest that the emergence of these "snakes" was probably genuine, as a superior strategy in this robot framework for this task in this environment, and not a mere result of the search space bias alone. Rolling can be an efficient gait as long as the environment allows for it, and similar behavior has been observed in other studies [32]. Undoubtedly, nonetheless, nature provides a plethora of examples where the complexity, in terms of limbs and utilized gaits, is much higher than the one shown in the example above. Therefore, we hypothesised that this complexity must be due to environmental constraints, that is, that different environmental pressures may lead to different morphologies and locomotion gaits. We verified this hypothesis [16], demonstrating that in our artificial life system different environments indeed can induce populations to different traits. However, we also showed that this biological notion should not be taken for granted when in artificial life systems, by demonstrating one example of severe environmental change that did not reflect in any significant changes neither in morphology or behavior. A possible reason for this case of [16] is that the task was too difficult. This might have led the search to a local optimum that did not induce the expected phenotypic differences while robots performed poorly on the task. This could indicate limitations in the encoding method, evolutionary algorithm or fitness function. In the present paper, we extend this work by performing a more general study of the effect of the environment beyond a specific setting. We evolve robots in two new environments, and compare their traits to robots evolved in the baseline environment (as in [16]). Furthermore, while in the previous work we assessed only behavioral and body morphological properties, now we also assess a controller property, that is, a brain property.

## 3 Methods

In our methodology, we use modular robots to represent the morphology (see Section 3.2) and neural networks to represent the controllers (Section 3.3). These two together represent the phenotypes, as they express the traits that ultimately, through the interaction with the environment, determine fitness. The evolutionary process acts on a higher level, the level of the genotypes, whose representation is explained in Section 3.4. Genotypes are converted into phenotypes through a mapping process, which is explained in Section 3.5. In the first generation, the genotype of the initial population is initialized according to the procedure described in Section 3.6. Each genotype is mapped into a phenotype and is subject to the evolutionary process explained in Section 3.9. During this evolutionary process, the operators of crossover and mutation are applied, which are explained respectively in Section 3.7 and Section 3.8.

### 3.1 Simulation

Our experiments were realized using a simulator called Gazebo, interfaced through a robot framework called Revolve [33].

### 3.2 Morphology

Each morphology phenotype (a 'body') is composed of modules [34] as shown in Fig 1. Each module has a cuboid shape, having slots where other modules can attach. The morphologies can only develop in 2 dimensions, that is, the modules do not allow attachment to the top or bottom slots, but only to the lateral ones. There are five different types of modules, as reported in Table 1: core components, bricks, vertical joints, horizontal joints, and touch sensors. Any module can be attached to any module through its slots, except for the touch sensors, which cannot be attached to joints. Each module type is represented by a distinct symbol (see

**Fig 1. On the left, the robot modules: Core-component with controller board (C).** Which is the head of the robot; Structural brick (B); Active hinges with servo motor joints in the vertical (A1) and horizontal (A2) axes; and Touch sensor (T). Modules C and B have attachment slots on their four lateral faces, and A1 and A2 have slots on their two opposite lateral faces; T has a single slot which can be attached to any slot of C or B. On the right, an example of simulated robot.

Table 1), and this is also the same language used in the genotype representation, described in Section 3.4.

### 3.3 Controller

A controller phenotype (a 'brain') is a hybrid artificial neural network (Fig 4, right), which we call Recurrent Central Pattern Generator Perceptron [16]. This network is formed by two types of nodes, that is, input nodes associated with the sensor modules, and oscillator neuron nodes associated with the joint modules. For every joint in the morphology, there exists a corresponding oscillator neuron in the network, whose activation function is defined by Eq (1), which represents a sine wave defined by amplitude, period, and phase offset parameters. This activation function adjusts the output to fit the range of our servo motors, as proposed in [33]:

$$O = 0.5 - \frac{a}{2} + \frac{\sin\left(\frac{2*\pi}{p} * (t - p * o)\right) + 1}{2} * a, \tag{1}$$

where, $t$ is the time step, $a$ is the amplitude, $p$ is the period, and $o$ is the phase offset. The parameters $a$, $p$, and $o$ can vary from 0 to 10.

The different oscillator neurons are not directly interconnected, and every oscillator neuron may or may not possess a direct recurrent connection.

Additionally, for every sensor in the morphology, there exists a corresponding input in the network, and each input might connect to one or more oscillator neurons. The oscillator neurons generate a constant pattern of movement, even if the robot is not sensing anything, so that the sensor inputs can be used either to reduce or to reinforce movements.

### 3.4 Representation

Our robot genotype is a generative model, and is represented with an L-System inspired in [35], conjointly encoding both morphology and controller. L-Systems are parallel rewriting systems [36] composed by a grammar defined as a tuple $G = (V, w, P)$, where

- $V$, the alphabet, is a set of symbols containing replaceable and non-replaceable symbols.

- $w$, the axiom, is a symbol from which the generative process starts.

- $P$ is a set of production-rules for the replaceable symbols, having one production-rule paired with each replaceable symbol.

**Table 1. Alphabet of the grammars.** Terminology is explained in Section 3.5.2.

| Modules | |
|---|---|
| **C** | core-component (axiom $w$) |
| **B** | brick |
| **A1($w_v, a_v, p_v, o_v$)** | vertical joint |
| **A2($w_h, a_h, p_h, o_h$)** | horizontal joint |
| **T($w_t$)** | touch sensor |

$w_v, w_h, w_t$ are sampled from a uniform distribution ranging from −1 to 1
$a_v, p_v, o_v, a_h, p_h, o_h$ are sampled from a uniform distribution ranging from 1 to 10

| Morphology-mounting commands | |
|---|---|
| **add_right** | add new module to the right of *module-reference* |
| **add_front** | add new module to the front *module-reference* |
| **add_left** | add new module to the left of *module-reference* |

| Morphology-moving commands | |
|---|---|
| **move_back** | move *module-reference* to the module at the back of *module-reference* |
| **move_right** | move *module-reference* to the module at the right of *module-reference* |
| **move_front** | move *module-reference* to the module at the front of *module-reference* |
| **move_left** | move *module-reference* to the module at the left of *module-reference* |

| Controller-moving commands | |
|---|---|
| **move_ref_I($t_i, d_i$)** | update *input-reference* with the input connected to edge $d_i$ of the neuron connected to edge $t_i$ of *input-reference* |
| **move_ref_N($t_n, d_n$)** | update *neuron-reference* with the neuron connected to edge $d_n$ of the input connected to edge $t_n$ of *neuron-reference* |

$t_i = \lceil\sqrt{v_1^2}\rceil$ and $t_n = \lceil\sqrt{v_2^2}\rceil$, and they are used to move the reference to a temporary node
$d_i = \lceil\sqrt{v_3^2}\rceil$ and $d_n = \lceil\sqrt{v_4^2}\rceil$, and they are used to move the reference to a definite node
$v_1, v_2, v_3, v_4$ are sampled from a normal distribution with $\mu = 0$ and $\sigma = 1$
If any of $t_i, d_i, t_n, d_n$ is greater than the number of edges of its corresponding node,
its value is updated with this number of edges.

| Controller-changing commands | |
|---|---|
| **add_edge($w_{e1}$)** | add an edge between *input-reference* and *neuron-reference* |
| **loop($w_l$)** | add a recurrent edge to *neuron-reference* |
| $w_{e1}, w_l$ are sampled from a uniform distribution ranging from −1 to 1 | |
| **mutate_edge($w_{e2}$)** | mutate the weight of the edge between *input-reference* and *neuron-reference* |
| **mutate_amp($m_a$)** | mutate amplitude of *neuron-reference* |
| **mutate_per($m_p$)** | mutate period of *neuron-reference* |
| **mutate_off($m_o$)** | mutate phase offset of *neuron-reference* $w_{e2}, m_a, m_p, m_o$ are sampled from a normal distribution with $\mu = 0$ and $\sigma = 1$ |

Each genotype is a distinct grammar, making use of the same alphabet (Table 1), and the alphabet is formed by symbols that represent types of morphological modules as well as commands for assembling modules together and others for defining the structure of the controller. The symbols in the category Modules are replaceable, while the symbols of all other categories are non-replaceable.

## 3.5 Genotype-phenotype mapping

The mapping from genotype to phenotype, that is, development, plays out in two stages that we call, respectively, *early* and *late* development.

**3.5.1 Mapping stage 1: Early-development.** The following didactic example depicts the process of rewriting of our L-System representing one possible genotype, that is, grammar. Here, the axiom of the grammar is rewritten into a more complex string of symbols according to the production-rules of the grammar. During the rewriting, for a number of iterations $k = 3$, each replaceable symbol is simultaneously replaced by the symbols of its production-rule.

Given the axiom $w = X$, the alphabet $V = \{X, Y, Z, a\}$ where the $a$ is the only non-replaceable symbol, and the production-rules $P = \{X: \{X, Y\}, Y: \{Z, a\}, Z: \{X, Z\}\}$, the rewriting follows as:

$$\text{Iteration } 0 : X$$
$$\text{Iteration } 1 : XY$$
$$\text{Iteration } 2 : XY\ Za$$
$$\text{Iteration } 3 : XY\ ZaXZa$$

The final string will contain non-repleaceble symbols (Modules) and repleaceble symbols (everything else). All these symbols can be interpreted with the process described hereafter.

**3.5.2 Mapping stage 2: Late-development.** The early-developed phenotype from stage 1 is an intermediate phenotype made as a string of symbols, which must be mapped (late-developed) into a final phenotype. To aid the process of construction of the late-developed phenotype, multiple positional references (turtles) are kept: a) a reference to the current module in the morphology, that we call a *module-reference*; b) a reference to the current oscillator neuron of the neural network of the controller, that we call a *neuron-reference*; c) a reference to the current sensor input of the neural network of the controller, that we call an *input-reference*; a reference to which slot of the current module a new module should be attached to, that we call a *slot-reference*.

From the beginning until the end of the string, each symbol is interpreted and developed. Nonetheless, for multiple reasons explained below, it is possible that a symbol ends up not being expressed in the phenotype. Furthermore, a maximum amount of $m$ modules is allowed in a morphology, so that during late-development, after reaching this maximum, any upcoming modules are not expressed in the phenotype. The late-development of the phenotype for morphology and controller is depicted in the fluxogram of Fig 2, and detailed hereafter, where we reference parts of this fluxogram through Roman numerals:

- ***I***: Because the first symbol of the string is always *C*, it is the first module to be added to the morphology, and the *module-reference* is updated with it. At this moment, the references of left, front, right, and back of the turtle are, respectively, left, up, right, and down (for a robot seen from top-down).

- ***II***: The interpretation of any Morphology-mounting command updates the *slot-reference* with the slot indicated by the command. If the *slot-reference* is not empty, it is overwritten, meaning that the command used for setting this previous slot into the reference is not expressed.

- ***III***: If the symbol is a module, it is coupled with the command in the *slot-reference* (if there is one).

- ***IV***: The addition of new modules requires both a Morphology-mounting command and a module. If the slot-reference is empty when interpreting a module, the module is not expressed in the phenotype, except for the *C* module, which is the very first module and needs no mounting command. When the *module-reference* is a joint, an attempt to attach it to the front slot is made, regardless of the mounting command. When the *module-reference* is the core-component, if its left, front, and right slots are occupied, an attempt to attach it to

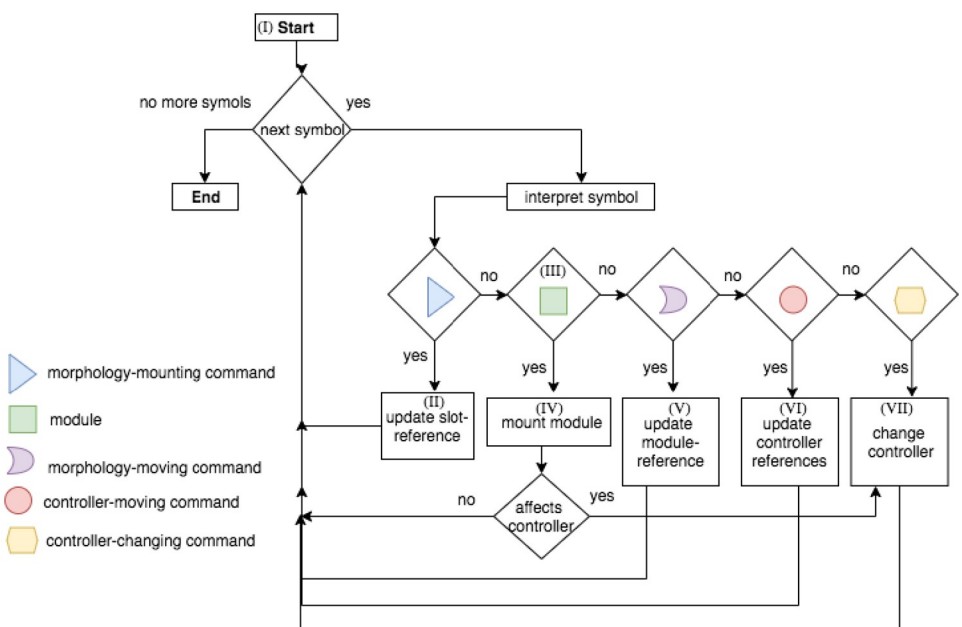

**Fig 2. Fluxogram of the late-development process.** From the left to right of the string, each symbol of the early-developed phenotype (string) goes thorough this process, being interpreted and developed (or not expressed).

the back slot is made, regardless of the mounting command. If the mounting attempt is made to a slot that is occupied, the module is not expressed, while the command remains in the *slot-reference*. If the newly mounted module intersects an existing one during the development, both the new module and its associated network node (if there is one) are not expressed. After mounting a new module, the *module-reference* remains in the parent module, and the *slot-reference* is emptied.

- **V**: The Morphology-moving commands update the *module-reference* according to the slot defined by the command. If the *module-reference* is a joint, any Morphology-moving command moves to the front slot.

- **VI**: The Controller-moving commands update the *neuron-reference* or *input-reference* according to the steps defined by the command, and is divided into two steps. The steps are illustrated by Fig 3.

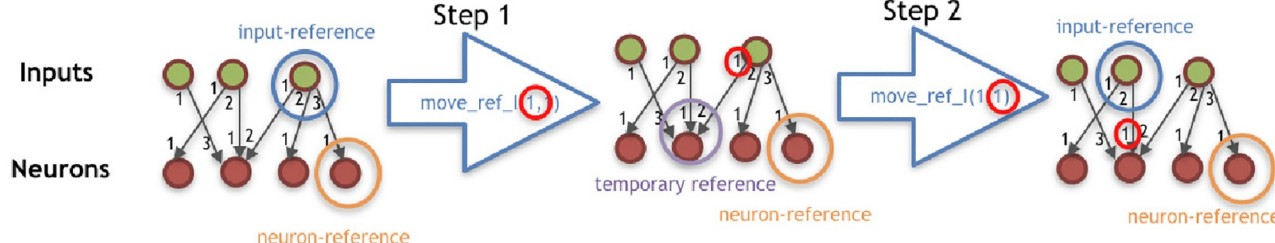

**Fig 3. Illustration of command move_ref_I($t_i$; $d_i$), having $t_i = 1$ and $d_i = 1$. The procedure of the command move_ref_N($t_n$; $d_n$) is analogous to this.**

- *VII*: The Controller-changing commands apply changes to the *neuron-reference* and/or *input-reference*, or to the edge connecting them. Controller-changing commands act upon the input and neuron nodes at the top (latest) of the stack. If there are no input or neuron nodes yet (according to the requirements of the command), the command is not expressed. If a newly mounted module is a joint, a new neuron is created possessing a connection weight that is drawn from a random uniform distribution between −1 and 1, and this neuron becomes the *neuron-reference*. When a new neuron is created, this generates an edge between this neuron and the *input-reference*. If there is no input yet, the neuron is stacked (oldest neuron remains as *neuron-reference*). If there is a stack of inputs, the new neuron is connected to all of them; for the edges, the input on the top of the list uses the weight possessed by the neuron, while all the other inputs in the stack use their own weights; finally, the stack is partially emptied keeping only the latest neuron, which becomes the *neuron-reference*.

  If a newly mounted module is a sensor, a new input is created possessing a connection weight that is drawn from a random uniform distribution between −1 and 1, and this input becomes the *input-reference*. When a new input is created, this generates an edge between this input and the *neuron-reference*. If there is no neuron yet, the input is stacked (the oldest input remains as *input-reference*). If there is a stack of neurons, the new input is connected to all of them; for the edges, the neuron on the top of the list uses the weight possessed by the input, while all the other neurons in the stack use their own weights; finally, the stack is partially emptied keeping only the latest input, which becomes the *input-reference*.

  For every new edge created from an input to a neuron, the edge is attributed a serial ID within the neuron. Analogously, for every new edge created from a neuron to an input, the edge is attributed a serial ID within the input.

An example of late-development is illustrated in Fig 4.

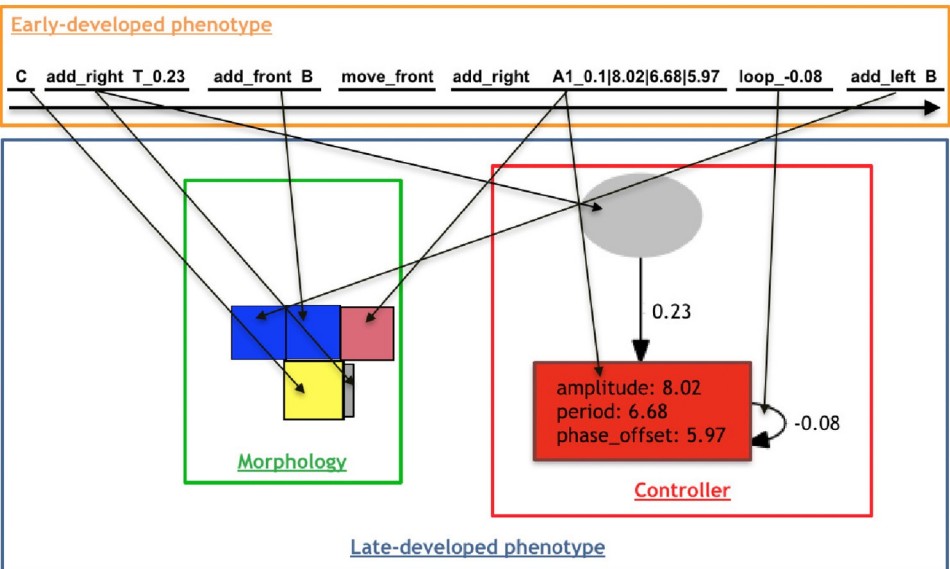

**Fig 4. Process of decoding an early-developed phenotype into a late-developed phenotype with morphology and controller.** From the left to right of the string, symbols are interpreted and developed, making incremental changes to the phenotype. An arrow going from the genotype to the phenotype should be interpreted as the process leading to the creation of the phenotype component pointed at by the arrow after the interpretation of the genotype component at the starting end of the arrow.

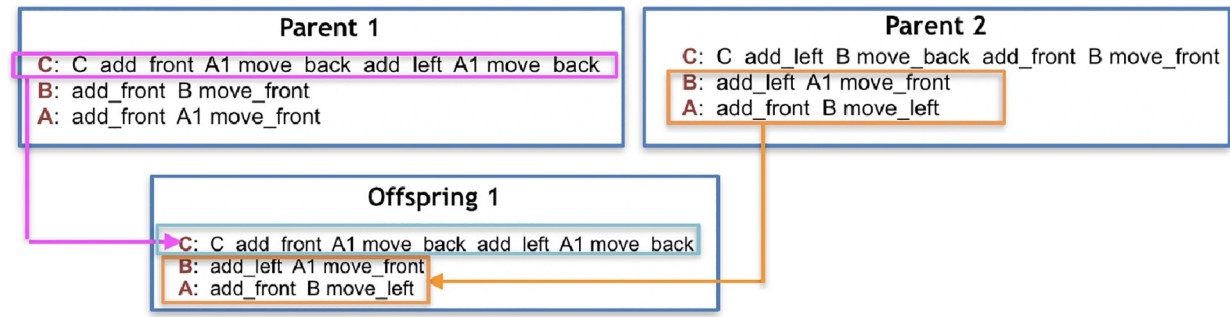

**Fig 5. a) and b) are examples of reproduction operators, and c) is an example of initialization using only 1 group of symbols for all cases of rules.**

### 3.6 Initialization

To initialize a genotype, for each production-rule, exactly one symbol is drawn uniformly random from each of the following categories in this order: Controller-moving commands, Controller-Changing commands, Morphology-mounting commands, Modules, Morphology-moving commands. This process is repeated $s$ times, where is $s$ sampled from a uniform random distribution ranging from 1 to $e$. This means that each rule can end up with 1 or maximally $e$ sequential groups of five symbols. The symbol C is reserved to be added exclusively (and surely) at the beginning of the production rule C. (Fig 5c).

### 3.7 Crossover

The crossovers are performed by taking complete production-rules uniformly at random from the parents (Fig 5a).

### 3.8 Mutation

Individuals are mutated by adding, deleting, or swapping one random symbol from a random production-rule in a random position (Fig 5b). As for the addition of symbols, all categories have an equal chance of being chosen to provide a symbol, and every symbol of the category also has an equal chance of being chosen. An exception is made to C to ensure that a robot has

one and only one core-component. This way, the symbol $C$ can not be added to any other production rules, neither removed or moved from the production rule of $C$. The operations adding, deleting, and swapping have an equal chance to happen. All symbols have the same chance of being removed or swapped.

### 3.9 Evolution

We are using overlapping generations with a population size $\mu = 100$. In each generation, $\lambda = 50$ offspring are produced by selecting 50 pairs of parents through binary tournaments (with replacement) and creating one child per pair by crossover and mutation. From the resulting set of $\mu$ parents plus $\lambda$ offspring, 100 individuals are selected for the next generation, also using binary tournaments. The evolutionary process is stopped after 100 generations, thus all together we perform 5050 fitness evaluations per run. For each environmental scenario, the experiment was repeated 20 times independently. A summary of the parameters for the evolutionary algorithm is provided in Table 2.

## 4 Experimental setup

The code needed to reproduce our experiments and analysis in available on GitHub https://github.com/ci-group/revolve/tree/a0a6496812cbec1208c3eb9fa4a0a21598ecb732/experiments/Environmental_influences_on_evolvable_robots. The resulting data is available in the supplementary material S1 Dataset, and also in the server ssh.data.vu.nl inside the kari-nemiras-plosone directory.

### 4.1 Environments and fitness functions

We experimented with two different environments, which are a) Flat: it is a plane flat floor; b) Tilted: it is a plane floor tilted in 5 degrees. The environments are depicted in Fig 6.

In the Flat and Tilted environments, the fitness function was defined by Eq (2):

$$f_1 = \begin{cases} s_x & \text{if } s_x > 0 \\ \frac{s_x}{10} & \text{if } s_x < 0 \\ -0.1 & \text{if } s_x = 0, \end{cases} \tag{2}$$

where $s_x$ is the speed of the robot as defined by Eq (4). This function measures the speed of the robots only in the $x$ axis, so to discourage robots to exploit locomotion in the $y$ axis, avoiding the proposed challenge of climbing the Tilted environment. Additionally, there are two

**Table 2. Parameters for the evolutionary algorithm.**

| | |
|---|---|
| Population size | 100 |
| Offspring size | 50 |
| Number of generations | 100 |
| Mutation probability | 80% |
| Crossover probability | 80% |
| Rewriting iterations $k$ | 3 |
| Maximum number of groups of symbols $e$ | 4 |
| Connections of the network range from | −1 to 1 |
| Oscillator parameters range from | 1 to 10 |
| Maximum amount of modules $m$ | 15 |
| Experiment repetitions | 20 |

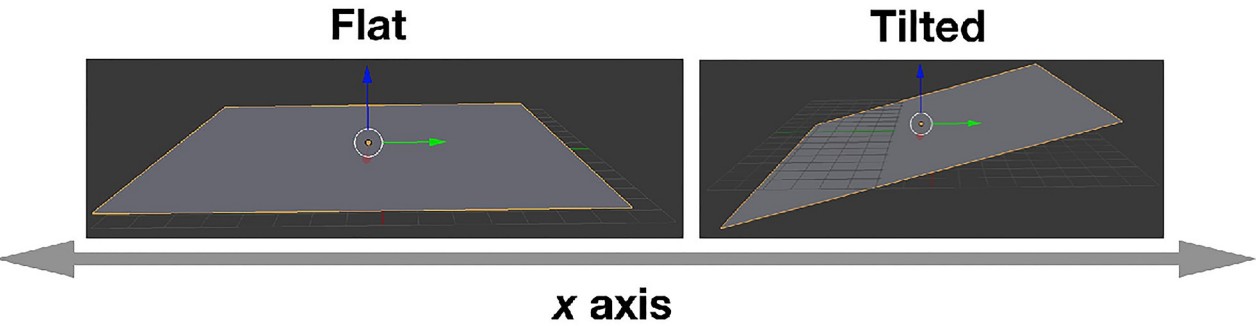

**Fig 6. The Flat and Tilted environments.**

penalties. The first penalty is the division by 10 used when the speed is negative, which aims at preventing that a "safe strategy" be much more beneficial than falling completely down the hill. This "safe strategy" is characterized by trying to avoid to fall too far from the starting point (due the effect of gravity), but without really climbing. The second penalty is the constant −0.1 used when speed is zero, which aims at disincentivizing robots that do not develop joints (and thus can not move) so to avoid the risk of falling.

## 4.2 Environmental conditions

We carried out two types of experiment using the same experimental setup, except for the environmental conditions in which the robots were evolved.

- *Static*: Robots live their whole lifetime in one same environment. Their lifetime, that is, simulation time, lasts 50 seconds.

- *Seasonal*: Robots live their lifetime across two different environments. They spend their first 50 seconds of lifetime in the Flat environment, and after that they spend 50 more seconds in the Tilted environment. Because in this case robots are evaluated in multiple environments, we treat this problem as multi-objective, where the fitness of each environment represents one of the objectives. The consolidation of these objectives into the final fitness is defined by Eq (3):

$$f_c = \sum_{i=1}^{n} d_i, \tag{3}$$

where in each generation, $d_i$ is the number of individuals in the population that are dominated by individual $i$, where individual is said to dominate another if it is better in both objectives. Importantly, all robots are evaluated in each environment regardless their performance in the other environment.

## 4.3 Robot descriptors

For quantitatively assessing morphological, control, and behavioral properties of the robots, we utilized a set of descriptors.

### 4.3.1 Behavioral descriptors.

1. **Speed**: Describes the speed (cm/s) of the robot along the *x* axis as defined by Eq (4):

$$s_x = \frac{e_x - b_x}{t}, \tag{4}$$

where $b_x$ is $x$ coordinate of the robot's center of mass at the beginning of the simulation, $e_x$ is $x$ coordinate of the robot's center of mass at the end of the simulation, and $t$ is the duration of the simulation.

2. **Balance**: We use the rotation of the head in the $x$–$y$ plane to define the balance of the robot. In general, the rotation of an object can be described in the dimensions roll, pitch, and yaw. We consider the pitch and roll of the robot head, expressed in degrees between 0 and 180 (because we do not care if the rotation is clockwise or anti-clockwise). Perfect Balance belongs to both pitch and roll being equal zero, so that the higher the Balance, the less rotated the head is. Formally, Balance is defined by Eq (5):

$$B = 1 - \frac{r + p}{t * 180 * 2},$$
(5)

where $r = \sum_{i=1}^{t} | r_i |$, representing the roll rotation accumulated over time, $p = \sum_{i=1}^{t} | p_i |$, representing the pitch rotation accumulated over time, and $t$ is the duration of the simulation.

**4.3.2 Morphological descriptors.** 1. **Size**: Total number $S$ of modules in the morphology.

2. **Proportion**: Describes the 2D ratio of the morphology and is defined with Eq (6):

$$P = \frac{p_s}{p_l},$$
(6)

where $p_s$ is the shortest side of the morphology, and $p_l$ is the longest side, after measuring both dimensions of length and width of the morphology (Fig 7).

3. **Relative Number of Limbs**: The number of extremities of a morphology relative to a practical limit. It is defined with Eq (7):

$$
\begin{aligned}
L &= \begin{cases} \dfrac{l}{l_{max}}, & \text{if } l_{max} > 0 \\[2ex] 0 & \text{otherwise} \end{cases} \\[3ex]
l_{max} &= \begin{cases} 2 * \lfloor \dfrac{(m-6)}{3} \rfloor + (m-6) \pmod 3 + 4, & \text{if } m >= 6 \\[2ex] m - 1 & \text{otherwise,} \end{cases}
\end{aligned}
$$
(7)

where $m$ is the total number of modules in the morphology, $l$ the number of modules which have only one face attached to another module (except for the core-component) and $l_{max}$ is the practical limit. This limit is the maximum amount of modules with only one face attached, that is, modules that represent a limb, which a morphology with $m$ modules could have if containing the same amount of modules arranged in a different way. This limit

(a) Proportion: 0.2 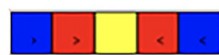   (b) Proportion: 1 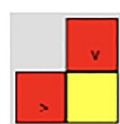

**Fig 7. Morphology (a) is disproportional and (b) is proportional.**

## (a) Limbs: 0.5 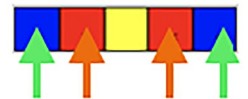   (b) Limbs: 1 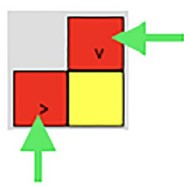

**Fig 8. Morphology (a) has four modules that could be extremities (considering the limit determined by the size of the morphology), but only the two indicated by green arrows are; (b) has the maximum number of extremities it could have.**

results logically from the nature of the possible connections in our system of Fig 8) presents examples.

4. **Relative Number of Joints**: This describes how movable the body is and is defined with Eq (8):

$$J = \begin{cases} \dfrac{j}{j_{max}}, & \text{if } m >= 3 \\ 0 & \text{otherwise,} \end{cases} \tag{8}$$

where $m$ is the total number of modules of the body, $j$ is the number of effective joints, that is, joints which have both of its opposite faces attached to the core-component or a brick, and $j_{max} = \lfloor (m - 1)/2 \rfloor$—the maximum amount of modules with two opposite faces attached that a body with $m$ modules could have, in an optimal arrangement (Fig 9).

5. **Symmetry**: This describes the reflexive symmetry of the body with Eq (9):

$$Z = \max(z_v, z_h), \tag{9}$$

where $z_h = o_h/q_h$—is the horizontal symmetry, and $z_v = o_v/q_v$—the vertical symmetry. For calculating each of these symmetry values, a referential center for the body is defined as the core-component. For both horizontal $h$ and vertical $v$ axes, a spine is determined as a line dividing the body into two parts according to the center and this axis. Each value is the number $o$ of modules that have a mirrored module on the other side of the spine (each match of modules accounts for two), divided by the total number $q$ of compared modules. The spine is not accounted for the comparison (Fig 10).

A complete search space analysis of the utilized robot framework and its descriptors is available in [28, 29], demonstrating the capacity of these descriptors to capture relevant robot properties, and proving that this search space allows high levels of diversity.

**4.3.3 Controller descriptor.** **Average period**: Describes the average (median) of the parameter Period among the oscillators of the controller (Fig 11). The higher this value, the slower the oscillation pattern, and thus slower the movement of the motors. It is defined with

## (a) Joints: 1 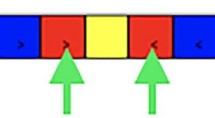   (b) Joints: 0.5 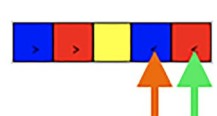

**Fig 9. Although both morphologies have two joints, in (b) the second joint is not effective, and would be only if the module indicated by the green arrow was switched with the one indicated by the orange arrow.**

**Fig 10. Morphology (a) has the modules indicated by green arrows horizontally reflected by the modules indicated by orange arrows; (b) has no modules reflected; (c) has the module indicated by the orange arrow vertically reflected by the modules indicated by the green arrow, but no reflection for the module indicated by the pink arrow.**

Eq (10):

$$D_{ap} = \frac{Md(P_l)}{m},$$

(10)

where $P_l$ is the set of all Period values of a robot controller defined as $P_l = \{p_l \ \forall \ l \in L\}$, and $m$ is the maximum value Period can assume, given that a controller has a set $L$ of oscillators defined as $l \in L$.

## 5 Results and discussion

In this section we analyze the effects of evolving robots under different environmental conditions on phenotypic and behavioral properties. We utilize two behavioral descriptors, five morphological descriptors, and one controller descriptor.

### 5.1 Static environmental conditions

Here we compare two populations of robots that evolved in the Static environmental condition, meaning they spent their whole life in one same environment. One population evolved in the Flat environment, while the other evolved in the Tilted environment. Differently from our previous work [16], the inclination in the Tilted environment is not 15 degrees, but 5. Still, we observe very similar effects of the inclination on the behavioral and morphological properties of the population when in comparison to the Flat environment.

We started by analyzing the effects of the environment on the behavior that is directly rewarded into the fitness function, that is, Speed. Through an initial intuition, we considered that the challenge of the task in the Tilted environment is greater than in the Flat environment, because its whole ground is (gradually) elevated, requiring a robot to climb it. Following this initial intuition, the level of difficulty of the locomotion task in Tilted is indeed higher than in Flat, since robots presented a Speed approximately three times lower (Fig 12) in the former.

Because Speed is the target behavior incorporated in our fitness function, we analyzed an extra behavioral descriptor, that is, Balance. Balance is not directly rewarded in the fitness but

**Fig 11. Example of controller.**

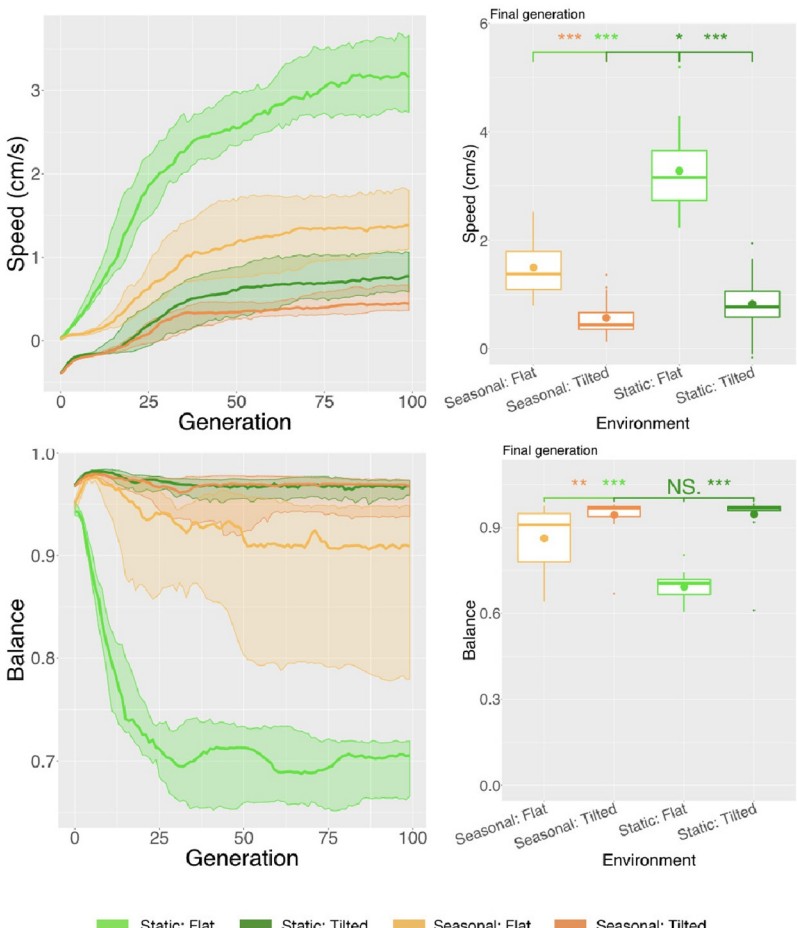

**Fig 12. Comparison of behavioral properties in different environmental conditions.** Line plots show the progression of the mean of the population (quartiles over all runs), while boxplots show the mean of the population in the final generation. Significance levels for the Wilcoxon tests in the boxplots are * < 0.05, ** < 0.01, *** < 0.001.

emerges to cope with the environment and the task. When the terrain in Tilted is inclined, while the terrain in Flat has no inclination, the natural state of non-moving robots would be rotated for the first, and non-rotated for the second. Nonetheless, the populations of robots evolved in Flat converged to unbalanced (rotated) robots, while the populations in Tilted converged to balanced (non-rotated) robots (Fig 12). Therefore, the convergence to these behaviors is not a mere artifact that any robot could achieve just by "being" in the environment, but also results directly from morphologies and controllers selected to cope with each environmental condition. The result of this behavioral descriptor agrees with the observed gaits, which in Flat are mostly rolling, while in Tilted mostly rowing or dragging. Apparently, "recklessly" rolling their bodies to boast away from the starting point is a good strategy when in a simple flat environment. On the other hand, maintaining a more stable rotation of their morphologies is more successful when the task concerns climbing. A video showing some of the evolved robots in each environment can be found in the supplementary material S1 Video, and also in the link https://youtu.be/HQcnmtMzb5U.

The previously observed differences in behavior are evidence of the effects of the environment on the robots. However, given that behavior is what emerges from the interaction among

morphology, controller, and environment, it is difficult to clearly separate how much of the behavior is an indirect result of the environment, and how much is a direct result. By indirect we mean its influence on changing the body and brain (hence, its behavior) evolutionarily, and by direct we mean its influence on the behavior in real-time during this interaction. Having such a challenge in mind, we additionally assessed a set of morphological properties, aiming to verify the indirect influence of the environment on the behavior. Fig 15 shows the progression of the average value for the morphological descriptors across the generations and the comparison of their average in the final generation. These charts show that the predominant morphological properties of the population of robots evolved in the Tilted environment are different from the ones evolved in the Flat environment, except for Symmetry. While in Flat evolution seems to be exploiting highly-actuated big disproportional morphologies, this strategy appears less suitable when the robots have to climb under the risk of falling down the hill with the help of gravity (Tilted), so they actually ended up smaller, more proportional, and with more limbs. The directional change for some of the curves of Tilted around generation 10 is due to a local optimum. It results from a strategy that seems to "assume it to be safer" to reduce movement than to risk falling during the climbing attempt. Therefore, in the early stages of evolution, the population converges to tiny robots that do not (or hardly) move. In later stages, it progresses for bigger robots, although these robots are smaller than the ones in Flat.

To better illustrate the differences between the properties of the final populations we plotted density maps with three example pairs of descriptors in Fig 13, allowing a multidimensional perspective. These charts show that the fittest robots evolved in Tilted spread to different areas of the space than those that evolved in Flat. Note that robots in Tilted have a higher Balance, a higher Rel. Number of Limbs, and a higher Proportion, demonstrating that it is hard to maintain a stable gait for climbing a hill when a robot has a single limb or when it is disproportionate. Robots in Flat instead do not possess many limbs, are disproportionate, and have a low Balance, because stability is less necessary to locomote on a plane flat floor. For visual inspection, Fig 14 shows the morphology of the best robot in the final generation of each run for both environments.

Finally, Fig 15 depicts the controller descriptor Average Period, but there was no significant difference between Flat and Tilted.

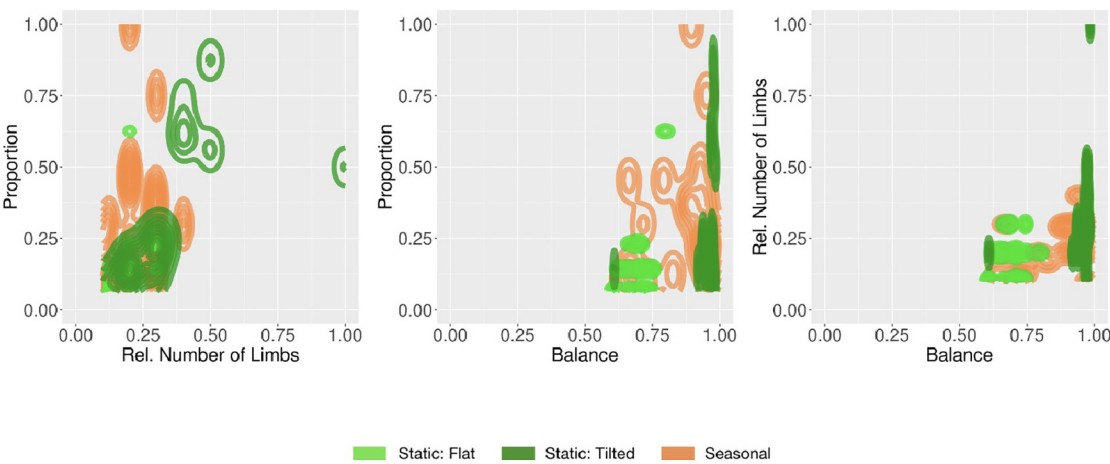

**Fig 13. Density maps for pairs of morphological descriptors in the final populations (all runs).**

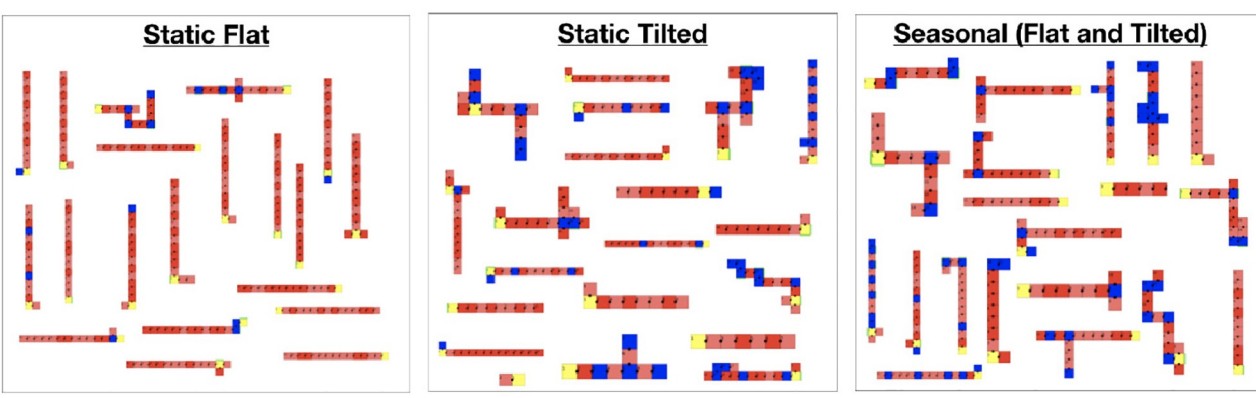

**Fig 14. Best robot of each experiment repetition in the different environmental conditions.**

## 5.2 Static versus Seasonal

Here we compare the two populations of robots presented in Section 5.1 to a new population. The new population was evolved in a Seasonal environmental condition, meaning that robots spent part of their lives in the Flat environment, and the remaining part in the Tilted environment. Importantly, this Seasonal setup can also be interpreted as robots having to perform multiple tasks and thus having multiple objectives. This is true because locomoting on a hill is a different task from locomoting on a flat terrain. In accordance to the "no free lunch" theorem, we expected a degradation on the Speed regarding at least one of the environments. In fact, because in the Seasonal environmental condition the search is trying to succeed in locomoting in both environments, this degradation indeed happened, and it took place for both environments. For both Flat and Tilted, the average Speed when evolving in Seasonal conditions was approximately half than when evolving in Static conditions (Fig 12). Interestingly, the Tilted environment, which proved more difficult as discussed in the previous section, degraded less severely than the Flat environment.

One probable cause for this is that, being a greater challenge, the Tilted environment applied a stronger selection pressure on the population. Another explanation is that the properties induced by the Tilted environment are more likely to generalise to both locomotion tasks than the ones induced by the Flat environment. This explanation is supported by experiments we presented in a previous paper [31], where we compared the robustness between populations evolved in the Flat environment and then tested in the Tilted environment (and vice versa). In this robustness test, we showed that robots evolved in Tilted could still perform locomotion to the side rewarded by the fitness function, while robots evolved in Flat and tested in Tilted were mostly falling down the hill.

In the Seasonal experiments, the median Speed is approximately three times lower in the Tilted than Flat. Although this is also true in the in the Static, the significance level obtained testing the difference between Seasonal Tilted and Static Tilted is ten times lower than the one obtained testing the difference between Seasonal Flat and Static Flat. Therefore, when in a Seasonal environmental condition, Speed is more similar to when in Static Tilted than when in Static Flat.

The emergent behavior measured with the Balance descriptor corroborates these observations. While Static Flat presents a much lower Balance than Static Tilted, this difference is less intense when comparing Seasonal Flat with Seasonal Tilted, while Seasonal Flat and Static Flat are very different. More importantly, Static Tilted and Seasonal Tilted present the same

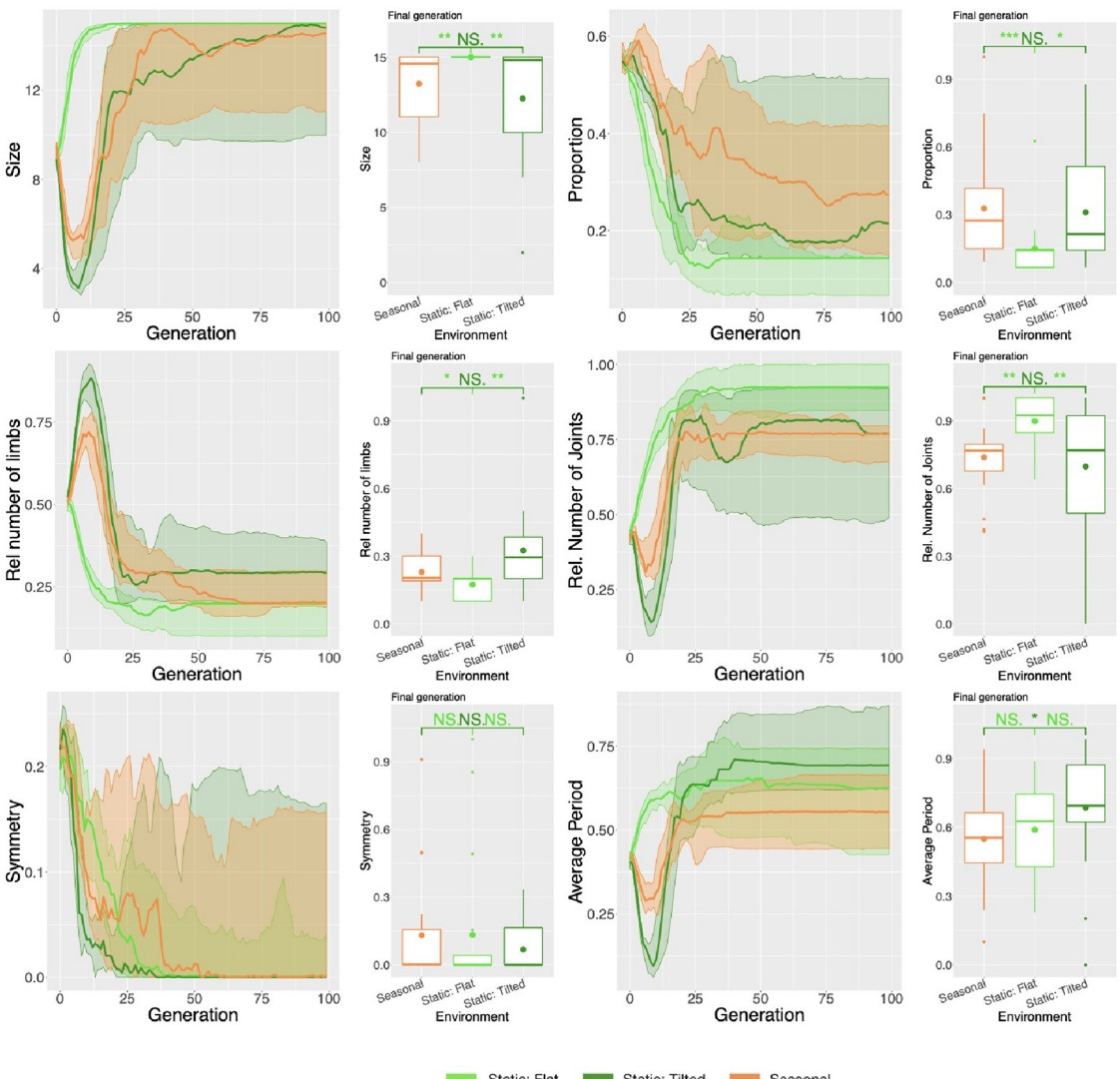

**Fig 15. Comparison of morphological properties in different environmental conditions.** Line plots show the progression of the mean of the population (quartiles over all runs), while boxplots show the mean of the population in the final generation. Significance levels for the Wilcoxon tests in the boxplots are $^* < 0.05$, $^{**} < 0.01$, $^{***} < 0.001$. Note that because phenotypic properties are the same in both environments for Seasonal, it is displayed only once in each chart.

Balance. Again, the behavior in Seasonal is more similar to the behavior in Static Tilted than to the behavior in Son-season Flat.

Beyond behavioral characteristics, the morphological properties exhibit this same dynamics. For every descriptor where Static Flat is different from Static Tilted, no difference exists between Static Tilted and Seasonal. One more time, in the Seasonal environmental condition, evolution favored the traits that are usually favored when evolving solely in the Tilted environment. Fig 14 illustrates how robots evolved in Seasonal resemble more robots evolved in Static Tilted than in Static Flat.

This result contrasts with a previous work [31] where we measured the effects of changing the environment throughout the evolutionary period. In that study, we started by evolving populations in the Flat environment, and in later generations, little by little we increased the

inclination of the floor towards 15 degrees. Interestingly, the traits of the populations that would normally emerge in the earliest stage, that is, the Flat environment, took over, even later on when evolving in the Tilted environment. Here, because we always evaluate every individual in both environments independently, the timing of when the environmental conditions take place does not matter. After all, the traits that took over were not the ones from Flat, but from Tilted. This possibly happened for the reasons we discussed above concerning difficulty and generalization.

Finally, we analyzed the Average Period, which is a property of the controller. Curiously and differently from all other descriptors, in this case, Seasonal and Static Tilted are different, while Seasonal is not different from Static Flat. In practice, this means that in Seasonal the movements of the motors happen faster than in Static Tilted, while not faster than in Static Flat. Meanwhile, it is not clear for us what this means. While morphological properties are intelligible and easily observable, controller properties seem less material and harder to interpret.

# 6 Conclusion

This paper studied the effects of diverse environmental conditions on behavioral, morphological, and controller properties of evolvable modular robots. We experimented with two environments: a) Flat: a plane flat floor; and b) Tilted: a plane floor tilted in 5 degrees. Our experiments investigated two types of environmental conditions. The first environmental condition was called Static, where an evolving population of robots spent its whole lifetime in the same environment. The second environmental condition was called Seasonal, where an evolving population of robots spent half of their lifetime in the Flat environment, and half in the Tilted environment.

Similarly to our previous work [16], also here we observed that, in the Static environmental conditions, each one of the environments created a different selection pressure to the populations. These selection pressures resulted in differentiation for behavioral and morphological properties, which we measured using several descriptors. Although evidence in natural systems would make these results look logical, previous work demonstrated that this dependence on the environment is very difficult to reproduce in an artificial evolutionary system. The paper [16] demonstrated an example of a drastic environmental change that did not induce any significant changes neither on morphology nor on behavior.

More importantly, in this paper we answered a new research question through the experiments with the Seasonal environmental condition. In this case, the emergent traits in the population gave in to the selection pressure existent in the Tilted environment, which is not only the most difficult one, but also the one that seems to induce a more general behavior for locomotion [31]. To more substantially define the implications of these findings, more research is certainly needed. For instance, if it is true that when having to deal with multiple environments the selection pressure of the most difficult environment will be the strongest pressure, then this should be taken into consideration when designing the maintenance of evolvable robot systems. Despite knowing that the environment is immensely determinant in any evolutionary process [6], experiments of Evolutionary Robotics that take the environment into consideration are very meager. Importantly, keeping the environment out of the investigatory loop could severely limit the conclusions of a great part of what has been experimented within the field. Therefore, the contribution of this work is a relevant stepping stone towards helping to increase the quality of artificial life systems, through trying to understand the influence of such a fundamental factor: the environment.

Furthermore, evolutionary robotics is not only concerned with creating utilitarian artificial life, that is, an autonomous robot system to perform some task, but is also useful for trying to understand how natural life evolved. While, knowing that the environment has a great influence on natural lifeforms [6], still little is known about how exactly it happens [37]. Controlling the environmental conditions, as we did in this paper, can be a useful tool to answer these questions through artificial evolution rather than with controlled lab experiments with natural evolution. Notably, while the natural evolutionary process is too slow to be experienced in a lab artificial evolution is much faster.

For future work, we propose to extend our encoding method with the capacity of phenotypic plasticity, that is, environmentally regulated phenotypes. This means that the encoding method will allow robots to develop morphologies and controllers, and thus behavior, according to the environmental conditions they face during their lifetime. Because in the current experiments robots demonstrated a degradation in the average performance when having to face multiple environments, we expect that our new encoding will help to reduce this degradation. Moreover, to increase the relevance of this investigation, more types of environments should be experimented with in the future. Finally, an analysis of the complexity [38] of the environments in relation to the complexity of the phenotypes would provide deeper insights into this subject.

## Supporting information

**S1 Video. Best robots video.**
(ZIP)

**S1 Appendix. Extra experiments.**
(PDF)

**S1 Dataset. Data from experiments.**
(ZIP)

## Acknowledgments

We would like to thank the ecologist Eleanor Collinson for helping with insights about the relevance of this research to ecology.

## Author Contributions

**Conceptualization:** Karine Miras, A. E. Eiben.

**Formal analysis:** Karine Miras.

**Investigation:** Karine Miras.

**Methodology:** Karine Miras, A. E. Eiben.

**Software:** Karine Miras.

**Supervision:** A. E. Eiben.

**Validation:** Eliseo Ferrante.

**Visualization:** Karine Miras.

**Writing – original draft:** Karine Miras.

**Writing – review & editing:** Eliseo Ferrante, A. E. Eiben.

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
