## [Decision Letter · Decision Letter 0]

17 Jan 2020

PONE-D-19-33498

Environmental influences on evolvable robots

PLOS ONE

Dear Mrs Miras,

Thank you for submitting your manuscript to PLOS ONE. After careful consideration, we feel that it has merit but does not fully meet PLOS ONE’s publication criteria as it currently stands. Therefore, we invite you to submit a revised version of the manuscript that addresses the points raised during the review process.

We would appreciate receiving your revised manuscript by Mar 02 2020 11:59PM. To enhance the reproducibility of your results, we recommend that if applicable you deposit your laboratory protocols in protocols.io, where a protocol can be assigned its own identifier (DOI) such that it can be cited independently in the future. For instructions see: http://journals.plos.org/plosone/s/submission-guidelines#loc-laboratory-protocols

We look forward to receiving your revised manuscript.

Kind regards,

Josh Bongard

Academic Editor

PLOS ONE

Journal Requirements:

2. Our internal editors have looked over your manuscript and determined that it is within the scope of our Open Soft Robotics Research Call for Papers. This collection of papers is headed by a team of Guest Editors for PLOS ONE: Guoying Gu (Shanghai Jiao Tong University), Aslan Miriyev (EMPA), Lucia Beccai (IIT), Matteo Cianchetti (Scuola Superiore Sant'Anna), Barbara Mazzolai (IIT) and Dana Damian (University of Sheffield).

The Collection will encompass a diverse range of research articles on soft robotics ranging from the development of future soft robots, durability, reliability, reproducibility and resilience in challenging environments. Additional information can be found on our announcement page: https://collections.plos.org/s/soft-robotics.

If you would like your manuscript to be considered for this collection, please let us know in your cover letter and we will ensure that your paper is treated as if you were responding to this call. If you would prefer to remove your manuscript from collection consideration, please specify this in the cover letter. Please note that inclusion in the call for papers will not result in a delay to publication and that the Guest Editors have final approval on what submissions are included.

4. We note that Figures in your submission contain copyrighted images. All PLOS content is published under the Creative Commons Attribution License (CC BY 4.0), which means that the manuscript, images, and Supporting Information files will be freely available online, and any third party is permitted to access, download, copy, distribute, and use these materials in any way, even commercially, with proper attribution. For more information, see our copyright guidelines: http://journals.plos.org/plosone/s/licenses-and-copyright.

1.    You may seek permission from the original copyright holder of Figures to publish the content specifically under the CC BY 4.0 license.

Additional Editor Comments (if provided):

Reviewers' comments:

Reviewer's Responses to Questions

**Comments to the Author**

1. Is the manuscript technically sound, and do the data support the conclusions?

Reviewer #1: Yes

Reviewer #2: Yes

2. Has the statistical analysis been performed appropriately and rigorously? 

Reviewer #1: Yes

Reviewer #2: Yes

3. Have the authors made all data underlying the findings in their manuscript fully available?

Reviewer #1: Yes

Reviewer #2: Yes

4. Is the manuscript presented in an intelligible fashion and written in standard English?

Reviewer #1: Yes

Reviewer #2: Yes

5. Review Comments to the Author

Reviewer #1: This work is a follow up work to [15], where the authors proposed a system based on L-Systems to evolve morphology + neural controllers for locomotion. In [15], they show that their approach can lead to diverse morphology when the environment is changed (i.e. tilting), but in some environments (i.e. obstacles) doesn't really lead to different robots. The main contributions in this work over [15] is addition of the Lava environment, and work on stimulating more diverse morphology.

While I really enjoyed reaching [15], honestly I believe this is an incremental work to their earlier work published at GECCO2019 [15], which is a solid paper, and a well-received work at the conference. It is my first time reviewing for PLOS so I'm not sure how it is like, but if this work were to be published at GECCO2020, and I was a reviewer, it would not have enough novelty. But sometimes it is okay for journals to published a polished version of conference papers with minor novelty contributions, so I think in that case it is okay to publish, so this I would like to leave it to the editors to decide. For this reason, I put in "major revision required" but if the editor is fine with publishing a polished version of a fantastic conference paper, then it should be "accept, or minor revision".

If more novelty is needed, I have a few ideas that might help make the work better, which moves in the direction of having less hand-engineering, and more ALIFE style emergent behaviors, which should lead to more interesting and diverse results:

- Rather than hand-engineering the reward function, think about using one based on survival, and looking at performance of the agents that "survived"? (i.e. not fall over or melt in lava)

- I noticed the sinusoidal component is hand-engineered into a locomotion, which feels unsatisfying given that walking with a cyclical motion can be an emergent skill (if sinusoidal gate is part of the search space of the neural controller), rather than hand-engineered feature.

- Perhaps thinking about having a population interact with each other in a single environment fighting for survival, and see what behaviors and morphologies occur in "self-play" or "survival of fittest" scenarios.

- the environments are not super interesting, to be honest. how about adding rewards like food, giving agents some perception (can be pixel perception of the env, or just distance and direction to nearby items), and see if the agents make use of their sensory inputs, to make things more life like?

[15] https://www.cs.vu.nl/~gusz/papers/2019-Effects_of_environmental_conditions_on_evolved_robot_morphologies_and_behavior.pdf

Reviewer #2: Overall, the work does provide promise for future evolutionary robotics. The methodology is sound and does collect substantial data from previous work. However, the broader implications of the experiments seem limited despite the amount of work that is referred to in the paper. Specifically, there are three main points that need attention and require revisions.

First, a clearly stated research objective (whole point of the research – i.e. why these experiments were done at all) and contribution of the results to the state of the art in evolutionary robotics (introduction) and referral to the objective and results significance in the discussion and conclusion sections.

Second, some discussion of environmental versus evolved morphological complexity is needed (preferably with reference to complexity measures of the environment and creature morphology), and the relationship of this papers results to that of the work presented in [16].

Third, an analysis and discussion of the reasons why results differed for the experiments presented in this paper versus those presented in the authors previous work ([15]).

Each of these points (together with suggestions for minor revisions) are elaborated upon in the following. However, a significant re-write of the introduction, discussion and conclusions sections are needed before this paper can be accepted for publication in this journal.

General comments and corrections:

Remove figure 1 – such a standard evolutionary process is well known, will be familiar to most readers – and is not necessary to reproduce here – rather simply refer to Eiben and Smith (2003).

Do not begin new sections so close to the end of page – e.g. section 3.2 and 3.5.

Be sure all figures are placed at the top of each page – e.g. figure 3, 4, 5 and 6.

Place links to all videos and ancillary material in footnotes.

Do not use “/” as a joiner for two words – e.g. “task/environment” – rather say: “task and environment, or both”.

The function of periods in oscillators in section 4.2.3 needs to be clarified – this section should be expanded a bit for a clearer explanation – as from the example in figure 12 – it is unclear what the oscillators are and what function the Period parameter has in relation.

A wider range of test environments (with an information-theoretic measure of environmental complexity – e.g. from low to high complexity environments) will be needed if they wish to make this research at all relevant to the evolution of morphological versus environmental complexity in nature (as the authors claim in the conclusions).

Results and Discussion.

With respect to the following – a measure of environmental complexity would greatly assist in classifying gradations of environments – as well as comparatively (evolved) morphological complexity – and thus give more weight to the results discussion (and conclusions drawn from this discussion).

“Through an initial intuition, we considered that the challenge of the task in the

Tilted environment is greater than in the Flat environment, because its whole ground is

(gradually) elevated, requiring a robot to climb it. Following this initial intuition, the

level of difficulty of the locomotion task in Tilted is indeed higher than in Flat, once

robots presented significantly lower speed (Fig 18) in the former.”

Conclusion

“dependence from the environment”  “dependence on the environment”

The following can be removed – rather simply say that observing natural evolution in controlled biological systems is usually not feasible due to the extended time periods of observing N generations and the simple nature (and thus limited morphological versus environment insights) of organisms with short generational periods.

“On the one hand, the natural evolutionary process is too slow to be experienced in a lab, and for most species, no human could experience a significant evolutionary period in her life span. Because of that, evolutionary experimentation is often done with a few specific species that have a short 520

life cycle and rapid evolution, for instance the fruit fly Drosophila subobscura [37], limiting the scope of what can be investigated. On the other hand, artificial evolution is much faster, i.e., experiments can be run in days or weeks.”

Also, the following is rather obvious and can be removed:

“Importantly, when we aim at achieving levels of complexity like the ones we observe in nature, it might be essential to reproduce, at least some of, the conditions which allowed nature to achieve these levels.”

Despite the following being mentioned a few times – there was no discussion or analysis as to why this paper yielded different results in the evolved behavior-morphology couplings.

“The paper [15] demonstrated an example of a drastic environmental change that did not induce any significant changes neither on morphology nor on behavior.”

Introduction

The research goal is too vague and non-specific as to what is being evaluated and what the potential overall contributions of such experimental evaluations.

Most importantly – the reason why the authors are doing these experiments at all is missing – in the conclusions it is stated that the role of evolutionary robotics (and assumedly experiments such as these) is to elucidate mechanisms of evolution leading to varied morphological complexities across environments) – however the possibility to explain anything about natural evolution from the results presented is way too tenuous. That leaves us with creating new designs for utilitarian robots to solve various tasks across various environments (also mentioned in the conclusion). However, this is not stated as a goal in the introduction – and the test environments are too far removed from any real-world environment (and the task is simply gait-control) – so that leaves the reader wondering what the actual objective (and thus contribution) of the experiments is exactly.

“In the current paper, we extend our previous work [15], and our main goal is to gain 47

deeper insights into the effect of the environment on a robotic evolutionary process. Our 48

experiments demonstrate and discuss the effects that diverse environmental conditions 49

can have on behavioral and phenotypic properties of evolvable populations of robots. 50

To this end, we follow an experimental approach. In summary, we consider three 51

different environments, define quantitative descriptors of behavioral, morphological, and 52

controller properties, and compare how they depend on the environments.”

The broader implications of the results presented in figure 13 need to discussed a bit more and the contributions of such results clearly stated.

“To better illustrate the differences between the properties of the final populations we plotted density maps with three example pairs of descriptors in Fig 13, allowing a multidimensional perspective. These charts show that the fittest robots evolved in Lava spread to different areas of the space than those that evolved in Flat.”

Also, further insights into the relationship between the environment and morphological evolution could be gained from changing the environment type mid-way through evolution, so as (for example) to elucidate the evolutionary conditions for generalists (morphological-function over multiple niches) versus specialists (morphological-function in a specific niche).

Also, testing the robustness of evolved coupled behavior, with respect to slightly deformed morphologies across a range of environments could also potentially yield more useful insights about the relationship between evolved behavior-morphologies and the environment.

However, this would need to be the subject of more extensive experiments – which do not necessary have to be implemented for this paper’s revisions.

Furthermore, while the importance of complexity is mentioned in the abstract and introduction, as well as previous work reviewed ([16]), and the authors seem to be testing the impact of increasingly complex environments on morphological evolution – there is no mention (or measure) of evolved morphological complexity – there was the Relative Number of Limbs metric (which could be construed as some form of complexity) – but results did not seem to present any significant difference for this metric (section 5.1).

At any rate, there needs to be some discussion about evolved morphological complexity relative to environmental complexity and the relationship of the results presented in this article to previous work ([16]) – this is especially important as this is the only other major work that varies the environment to test varying morphological evolution.

Other corrections:

# Page Line

16 434-436

Text does not match figure

Text states that “In Fig 15 we see that the average period in the alternative environment (Lava) is significantly higher than in the baseline environment (Flat).” However, the figure shows the difference between environments as not significant. It appears the wrong figure may have been referenced.

# Page Line

2 33-36

Imprecise text. It is not clear why the research in [12,13,14] does not constitute “substantial

experimental evidence”.

3 66-69

Imprecise text. It may be better to state clearly that the environment did not change from a flat

plane in [25].

3 72-75

Imprecise text. It seems worth mentioning that there was evidence of differences in morphological

traits (legged locomotion), but these were not measured or investigated in depth.

3 90

Unexplained terminology. The use of the term “modules” is unclear without the context of the later chapters in this paper which describe what "modules" are.

3 90

Unclear description. The description of the shape of the robot is difficult to understand without the

context of the later chapters in this paper which describe the robot modules.

3 95

Imprecise text

It is unclear whether “task/environment” means task “task and environment”, “task or environment”, or “task and/or environment”

3 110

Correction “from” should be “for”

4 143

Unexpanded acronym. First use of the acronym CPG has not been expanded to full words.

5 Eq. 1

Require more explanation. It would be useful to the reader to know what informed the form of this modified sine wave.

9 257

Correction “wights” should be “weights”

9 260-261

Correction “id” should be capitalised as “ID”

9 Fig 4.

Correction Text in arrows should be “move_ref_I(1,1)” instead of “move_ref_S(1,1)”

10 268

Correction “…being s…” should be “…where s is sampled…”

11 290

Correction “5.050” should be “5050”

12 310-312

Run-on sentence. The phrasing of this sentence is difficult to follow due to complex construction and

use of commas.

13 337

Unclear. It is not clear what constitutes the head of the robot.

13 339

Introduction of shorthand text.

 and should be introduced as shorthand for pitch and roll.

13 354/Eq. 8

Require further explanation. It is not obvious what the practical limit is or how it was determined.

14 383

Require further explanation. It is not obvious what and are in the equation.

18 448

Correction “once” should be “since”

6. PLOS authors have the option to publish the peer review history of their article (what does this mean?). If published, this will include your full peer review and any attached files.

Reviewer #1: No

Reviewer #2: No

---

## [Author Response · Author response to Decision Letter 0]

11 Feb 2020

Thank you very much for all the feedback. We applied all due changes and they are explained in the rebuttal letter. In summary, we realised new experiments/analysis to increase the novelty of the paper, and also clarified our objectives and contributions.

---

## [Decision Letter · Decision Letter 1]

29 Apr 2020

PONE-D-19-33498R1

Environmental influences on evolvable robots

PLOS ONE

Dear Mrs Miras,

Thank you for submitting your manuscript to PLOS ONE. After careful consideration, we feel that it has merit but does not fully meet PLOS ONE’s publication criteria as it currently stands. Therefore, we invite you to submit a revised version of the manuscript that addresses the points raised during the review process.

I am very sorry for the decision-making process, which has been too long. This was partly due to the pandemic and its impact on the functioning of universities and the capacity of researchers who are also involved in teaching and e-learning. I have given the manuscript a minor revision in terms of small changes needed based on the review. But after their implementation, nothing will prevent the process of rapid acceptance of the manuscript.

We would appreciate receiving your revised manuscript by Jun 13 2020 11:59PM. To enhance the reproducibility of your results, we recommend that if applicable you deposit your laboratory protocols in protocols.io, where a protocol can be assigned its own identifier (DOI) such that it can be cited independently in the future. For instructions see: http://journals.plos.org/plosone/s/submission-guidelines#loc-laboratory-protocols

We look forward to receiving your revised manuscript.

Kind regards,

Denis Horvath

Academic Editor

PLOS ONE

Reviewers' comments:

Reviewer's Responses to Questions

**Comments to the Author**

1. If the authors have adequately addressed your comments raised in a previous round of review and you feel that this manuscript is now acceptable for publication, you may indicate that here to bypass the “Comments to the Author” section, enter your conflict of interest statement in the “Confidential to Editor” section, and submit your "Accept" recommendation.

Reviewer #2: All comments have been addressed

2. Is the manuscript technically sound, and do the data support the conclusions?

Reviewer #2: Yes

3. Has the statistical analysis been performed appropriately and rigorously? 

Reviewer #2: Yes

4. Have the authors made all data underlying the findings in their manuscript fully available?

Reviewer #2: No

5. Is the manuscript presented in an intelligible fashion and written in standard English?

Reviewer #2: Yes

6. Review Comments to the Author

Reviewer #2: The following minor comments and corrections should be addressed by the authors before the manuscript can be accepted for publication.

---

Introduction

We hypothesize that it has to do with the relative simplicity of these environments compared to the richness of (environmental) factors that determine the selection probabilities in Nature.

“in Nature” -> “in nature”.

I would suggest rephrasing to just say “selection in nature” as natural selection may well be a bit more complex than a set of probabilities.

“Curiously, in these (unpublished) experiments the same type of robots evolved in very different environments.”

I assume these unpublished experiments were those previously done by the authors? – If so I would recommend placing a footnote here – with a link to the results (ideally graphs) on an online repository – so as interested readers can look at them for reference and comparison.

3.8 Mutation

“adding/deleting/swapping” -> adding, deleting or swapping

“production-rule/position” -> production-rule/position

Also, fix in other sections: / should be replaced by “,” and “or” (as above).

Throughout the article, make sure there are no sentences split between pages – e.g.

“into two steps. The steps are illustrated by Fig 3.” On page 9.

Also, throughout the article, replace: “i.e.” -> “that is,” and “e.g.” -> “for example,”

Conclusion

“they face during their life.” -> they face during their lifetime.

“Having in mind that in the current experiments robots presented a degradation in the average”

->

“In the current experiments robots demonstrated a degradation in the average”

7. PLOS authors have the option to publish the peer review history of their article (what does this mean?). If published, this will include your full peer review and any attached files.

Reviewer #2: No

---

## [Author Response · Author response to Decision Letter 1]

30 Apr 2020

Thank you for the review. All comments were followed and are explained in the letter.

---

## [Editor Report · Decision Letter 2]

14 May 2020

Environmental influences on evolvable robots

PONE-D-19-33498R2

Dear Dr. Miras,

We are pleased to inform you that your manuscript has been judged scientifically suitable for publication and will be formally accepted for publication once it complies with all outstanding technical requirements.

With kind regards,

Denis Horvath

Academic Editor

PLOS ONE
---

## [Editor Report · Acceptance letter]

18 May 2020

PONE-D-19-33498R2 

Environmental influences on evolvable robots 

Dear Dr. Miras:

I am pleased to inform you that your manuscript has been deemed suitable for publication in PLOS ONE. Congratulations! Your manuscript is now with our production department. 

With kind regards,

on behalf of

Dr. Denis Horvath 

Academic Editor

PLOS ONE